# Cognitive and Behavioral Outcome of Pediatric Low-Grade Central Nervous System Tumors Treated Only with Surgery: A Single Center Experience

**DOI:** 10.3390/diagnostics13091568

**Published:** 2023-04-27

**Authors:** Matilde Taddei, Silvia Esposito, Gianluca Marucci, Alessandra Erbetta, Paolo Ferroli, Laura Grazia Valentini, Chiara Pantaleoni, Stefano D’Arrigo, Veronica Saletti, Bianca Pollo, Rosina Paterra, Daria Riva, Sara Bulgheroni

**Affiliations:** 1Department of Paediatric Neuroscience, Fondazione IRCCS Istituto Neurologico Carlo Besta, 20133 Milan, Italy; 2Neuropathology Unit, Fondazione IRCCS Istituto Neurologico Carlo Besta, 20133 Milan, Italy; 3Department of Neuroradiology, Fondazione IRCCS Istituto Neurologico Carlo Besta, 20133 Milan, Italy; 4Department of Neurosurgery, Fondazione IRCCS Istituto Neurologico Carlo Besta, 20133 Milan, Italy; 5Molecular Neuroncology Unit, IRCCS Carlo Besta, 20133 Milan, Italy

**Keywords:** neurocognitive outcome, behavioral disorders, low-grade brain tumor, children

## Abstract

Background: The present mono-institutional report aimed to describe the cognitive and behavioral outcomes of low-grade central nervous system (CNS) tumors in a cohort of children treated exclusively with surgical intervention. Methods: Medical records from 2000–2020 were retrospectively analyzed. We included 38 children (mean age at first evaluation 8 years and 3 months, 16 females) who had undergone presurgical cognitive–behavioral evaluation and/or at least 6 months follow-up. Exclusion criteria were a history of traumatic brain injury, stroke, cerebral palsy or cancer-predisposing syndromes. Results: The sample presented cognitive abilities and behavioral functioning in the normal range, with weaknesses in verbal working memory and processing speed. The obtained results suggest that cognitive and behavioral functioning is related to pre-treatment variables (younger age at symptoms’ onset, glioneuronal histological type, cortical location with preoperative seizures), timing of surgery and seizure control after surgery, and is stable when controlling for a preoperative cognitive and behavioral baseline. Younger age at onset is confirmed as a particular vulnerability in determining cognitive sequelae, and children at older ages or at longer postsurgical follow-up are at higher risk for developing behavioral disturbances. Conclusions: Timely treatment is an important factor influencing the global outcome and daily functioning of the patients. Preoperative and regular postsurgical cognitive and behavioral assessment, also several years after surgery, should be included in standard clinical practices.

## 1. Introduction

Consensus has been reached among clinicians and researchers that children presenting central nervous system (CNS) tumors are at higher risk for neurocognitive impairment and behavioral complication of degrees varying in accordance with clinical variables [1,2]. 

Although children with brain tumors are in general described as having global cognitive and adaptive functioning within average ranges [2,3], and high-grade tumors have been associated with worse functional outcomes [4,5,6], the presence of low-grade tumors still represents a risk for developing intellectual, neuropsychological and behavioral disturbances [7]. These may be caused by the neurological disease itself or secondarily by the treatment (surgery, chemotherapy, radiation) [5,8]. Moreover, children with low-grade tumors might have a longer clinical history before the diagnosis due to the slow-growing nature of the tumors and later onset of symptoms and signs (symptoms related to intracranial pressure, seizures, movement and oculomotor disorders) compared to patients with high-grade tumors, and this may increase the rate of pathological findings in neurocognition [9]. To date, some studies have shown that clinical factors influencing the neurocognitive prognosis of children with low-grade tumors encompass types of treatment [8], tumor size and location [4,10,11], the presence of pre-existing neurological disturbances and cancer predisposing syndrome [10,11] and to some extent younger age at diagnosis [12].

Repeated standardized testing of cognitive, behavioral and adaptive functioning for longitudinal control is recommended to understand the progress of neurocognitive sequelae and is often guaranteed in clinical practice [7], although drop-out risk or incomplete testing often limit data collection for research purposes [4,7]. Moreover, most patients are tested post-treatment, while cognitive and behavioral evaluation after diagnosis, but prior to surgery, is relatively absent, although it is important for establishing neurocognitive baselines and understanding what problems may pre-exist [4]. Case reports and longitudinal studies have demonstrated the feasibility of presurgical assessment to establish baseline functioning [4,6,9,13,14], and longitudinal assessment to disentangle the influence of clinical factors, such as tumor location and aggressiveness, from that of treatment on cognitive outcomes [4,6,10]. 

Surgery has become the elective treatment of choice for low-grade tumors occurring in childhood [15], and clinical practice consensus highlights the importance of monitoring the global intellective and behavioral functioning of patients [7]. Gross total resection is considered the most consistent prognostic factor for progression-free survival [15], but the relation between residual tumor after surgery and neuropsychological outcome is less clear [10,11,12]. To date, some studies have described neuropsychological sequelae of pediatric low-grade brain tumors in children treated only with surgery [6,13,16,17,18] or compared patients treated only by surgery with those treated with radio and chemotherapy [5,19]. A few of these studies have identified some predictors of lower postsurgical cognitive outcome, such as tumor lateralization [16], larger tumor size, supratentorial tumor location and history of seizures [18], but only post-treatment evaluation was analyzed. Two reports also provided data about preoperative evaluation of children with posterior fossa pilocytic astrocytoma, evidencing that some degree of neuropsychological impairment might be present already at the time of diagnosis and that preoperative factors, such as the baseline functioning of the child and tumor location, influenced the postsurgical outcome more than the surgical treatment [6,13]. The authors generally excluded severe neurological diseases but did not collect extensive information about pre-existing conditions that could impact cognitive functioning such as cancer-predisposing syndromes. To the best of our knowledge, no data are available about comparisons between different low-grade histological subtypes and about the effects of other developmental variables such as age at symptoms’ onset in predicting cognitive or behavioral functioning.

Finally, although pediatric low-grade brain tumor survivors experience long-term psychological, behavioral and affective problems [20], their behavioral and emotional adaptation has been investigated to a lesser extent than their intellectual and neuropsychological functioning [21]. Studies comparing pre- and postsurgical emotional and behavioral functioning in the same subjects are scarce and limited to children with posterior fossa tumors, in which a favorable outcome of both internalizing and externalizing problems after surgical interventions is described [6,13].

The present study aimed to describe the cognitive and behavioral outcomes of children with low-grade CNS tumors treated only with surgery and to identify clinical features associated with intellective and behavioral functioning. We retrospectively analyzed medical records of the Pediatric Neuroscience Department of our institution regarding children and adolescents who underwent presurgical cognitive and behavioral evaluation and/or at least six months’ postsurgical follow-up. The presence of premorbid risk factors for neurocognitive and behavioral disturbances has been excluded. We were interested in investigating the roles of surgical timing, surgical outcome (residual tumor and seizure control) and tumor location, which are still not clear in the literature, and in providing initial data about specific tumor histological subtypes and age at symptoms’ onset. We also wanted to clarify the effect of surgical treatment in predicting cognitive/behavioral outcomes while controlling for the baseline preoperative functioning of the subjects by analyzing a group of children who underwent both pre- and postsurgical assessment.

## 2. Materials and Methods

### 2.1. Participants

The sample was selected after a retrospective analysis of medical records from a 20-year period (2000–2020) about children and adolescents attending a clinical setting specialized in the care of pediatric brain tumors of the Fondazione Istituto di Ricovero e Cura a Carattere Scientifico (IRCCS) Istituto Neurologico Carlo Besta in Milan. From the internal cancer registry of the institute we selected children and adolescents aged 2 to 16 years who underwent presurgical cognitive and behavioral evaluation and/or at least 6 months’ postsurgical follow-up. Exclusion criteria were premorbid history of traumatic brain injury, stroke, cerebral palsy, neurofibromatosis type 1 (NF1) or other cancer-predisposing syndromes, as well as being treated with multiple surgical treatments or other adjuvant therapies. 

In total, 575 children and adolescents with low-grade brain tumors visited our institute between 2000 and 2020; 193 were treated with surgery in the same period, and 61 out of the surgically treated total sample were evaluated at the neuropsychology service. Out of these 61 patients, 23 were excluded due to the following conditions: non-valid neuropsychological evaluation (incomplete, within 6 months from the intervention), n = 10; presence of tumor-predisposing syndromes or other premorbid history of neurological disorders, n = 8; and multiple surgical treatments or adjuvant therapies, n = 5. The selection resulted in a final sample of 38 children. 

We collected information about low-grade tumor types based on specific biomolecular markers, tumor location, age at symptoms’ onset, age at diagnosis (i.e., first diagnostic radiological suggestion) and intervention, surgical outcome and pre- and postsurgical symptoms. 

Tumors were diagnosed and classified by neuropathologists with expertise in CNS tumors, according to the 2021 WHO classification system (CNS5) [22]. An immunohistochemical work-up included the identification of glial and neuronal markers (e.g., GFAP, Olig 2, IDH1, p53, synaptophysin, MAP-2, SMI32, CD34, etc.). In a few cases, assessment of molecular BRAF mutation status had been performed. 

An expert neuroradiologist reviewed both baseline and follow-up brain MRI scans to define tumors’ localization and the extent of surgical resection.

All the subjects’ parents gave their informed consent for personal data use for scientific aims before they participated in the study. The study was conducted in accordance with the Declaration of Helsinki and was approved by the Ethics Committee of the Fondazione IRCCS Istituto Neurologico Carlo Besta.

### 2.2. Cognitive Evaluation

Pre- and postsurgical cognitive evaluation encompassed the administration of developmental scales for infant development or intelligence scales. Half of the patients underwent attention, executive function, memory, visuo-spatial or language evaluation in the postsurgical period, but this specific neuropsychological assessment is not part of the present study. All subjects were assessed in a quiet room by examiners who had specific training on child assessment. For school-age children, intelligence was evaluated using the Wechsler Intelligence Scale for Children-Revised (WISC-R), third (WISC-III) or fourth (WISC-IV) edition [23,24,25]. In six cases, presurgical evaluation of pre-school-age children was performed via the Wechsler Pre-school and Primary Scale of Intelligence-Revised [26], Wechsler Pre-school and Primary Scale of Intelligence-III [27] or Wechsler Pre-school and Primary Scale of Intelligence-IV [28]. All the above mentioned scales allow the generation full-scale IQ, verbal IQ and nonverbal quotient scores. Furthermore, the WISC-III, WPPSI-III, WISC-IV and WPPSI-IV also provide indices of working memory/freedom from distractibility and processing speed indices. In two cases, the intelligence evaluation in the presurgical period was conducted by administering the Colored Progressive Raven Matrices [29] due to difficulties in organizing a complete intelligence scale administration before the surgical operation. All the postsurgical evaluations were performed with Wechsler scales. 

Different versions of the Wechsler scales are composed of different subtests. In general, the verbal domain (VIQ) includes the assessment of verbal fluency and word knowledge (Vocabulary), concept formation and verbal abstract reasoning (Similarities), social knowledge, practical judgment in social situations (Comprehension), general cultural knowledge (Information), short-term verbal memory and attention (Digit-span), verbal sequencing abilities and verbal working memory (Letter-number sequencing) and mental arithmetic ability (Arithmetic). The nonverbal performance domain (PIQ) assesses visual–spatial visualization and analysis and nonverbal concept formation (Block design), perceptual reasoning (Matrix Reasoning), categorical reasoning (Picture Concepts), logical and sequential story organization (Picture arrangement), visual-motor integration speed (Coding), visual scanning and selective attention speed (Symbol search). For the purpose of the present study, we also analyzed two more factorial indices: working memory index (WM), also called freedom from distractibility (FD) in the WISC-III, investigating only verbal short-term and working memory, and processing speed index (PS), investigating only the ability to rapidly elaborate visual and graphic stimuli in timed paper–pencil tasks. The Wechsler scales’ raw scores were converted to age-corrected subtest standard scores that were normally distributed with a mean of 10 and an SD of 3. The sum of age-scaled scores was converted into an overall standard score with a mean of 100 and a standard deviation of 15.

In two cases, the presurgical evaluation of pre-school-age children was performed with the Griffiths Mental Scale for Development [30,31] to evaluate their psychomotor development level. For each scale, the raw scores were converted to age-corrected standard scores with a mean of 100 and a standard deviation of 15. A general quotient (GQ) and six subquotients were obtained. The Locomotor subscale assesses gross motor skills, the Personal–Social subscale measures proficiency in the activities of daily living and interaction with other children, the Language subscale measures expressive and receptive language, the Eye and Hand Co-ordination subscale focuses on fine motor skills, manual dexterity and visual monitoring skills, the Performance subscale assesses nonverbal reasoning, and the Practical Reasoning subscale assesses the ability to solve practical problems, understand basic mathematical concepts and moral issues.

### 2.3. Behavioral Assessment

The presence of behavioral problems was assessed by psychopathological questionnaires, namely, the CBCL versions for 6–18 years or 4–18 years, and the version for 1 ½ −5 years, respectively for school-age and pre-school-age children [32,33]. The questionnaires were completed in the pre- and/or postsurgical period by parents/caregivers of 36 out of the 38 children. 

The norm-referenced CBCL describes a child’s functioning during the previous six months (school-age forms) or three months (pre-school-age form). The items measure specific emotional and behavioral problems on a 3-point Likert scale (0 = “Not True”, 1 = “Somewhat or Sometimes True” and 2 = “Very True or Often True”). 

All the CBCL forms contain two empirically derived global scales and eight syndrome scales. The CBCL/6–18 y and 4–18 y versions’ global Internalizing domain (CBCL/Int) contains three syndrome scales: Anxious/Depressed, Withdrawn/Depressed and Somatic Complaints. The global Externalizing domain (CBCL/Ext) contains the Rule-Breaking Behavior and Aggressive Behavior syndrome scales. Three other syndrome scales do not belong to either broadband scale: Social problems, Thought problems and Attention problems. A Total problems scale (CBCL/Total) quantifies overall impairment and is derived from the raw score sum of all eight syndrome scales. On the CBCL/1 ½−5 y version, the CBCL/Int scale contains four syndromic scales (Emotionally Reactive, Anxious/Depressed, Somatic Complaints and Withdrawn). The CBCL/Ext domain contains the Attention Problems and Aggressive Behavior scales. One other syndromic scale does not belong to any Internalizing or Externalizing scale: Sleep Problems. CBCL/Total quantifies overall impairment and is derived from the raw score sum of all eight syndrome scales. Raw scores for each scale are converted to norm-referenced T-scores (M = 50, SD = 10). For the pre-school-age and school-age (only 6–18) versions of the CBCL, *Diagnostic and Statistical Manual of Mental Disorders* (DSM)-Oriented Scales are also present as a supplement the CBCL syndrome scales: Affective Problems, Anxiety Problems, Attention Deficit and Hyperactivity Disorder (ADHD) Problems and Oppositional Defiant Problems are common among the 6–18 y and 1 ½–5 y versions; the Somatic Problems and Conduct Problems scales are present only in the school-age CBCL, and the Pervasive Developmental Problems scale is present only in the pre-school-age CBCL. “Pathological” scores are indicated by T-scores ≥64 on the global scales, and ≥70 on the syndromic and DSM-Oriented scales. “Borderline” ranges are considered to be 60–63 and 65–69 on the global and syndromic scales, respectively.

### 2.4. Statistical Analysis

Statistical analyses have been implemented by using the SPSS Statistics 20 software [34]. One-way ANOVA and a *t*-test for independent samples were used to evaluate the influence of categorical clinical variables (e.g., tumor location, tumor lateralization, tumor type, surgical outcome, postsurgical seizure control) on pre- and postsurgical cognitive and behavioral functioning. Spearman correlation with Bonferroni correction for multiple comparison was used to evaluate the influence of continuous clinical variables such as age at symptoms’ onset, age at evaluation, age at intervention and time between symptoms’ onset/diagnosis and intervention on pre- and postsurgical cognitive and behavioral functioning. Repeated measure ANOVA and a *t*-test for paired samples were used to analyze within-subject profiles and to compare pre- and postsurgical outcomes on global, verbal and nonverbal intelligence or developmental quotients, as well as emotional–behavioral problems in children presenting both pre- and postsurgical evaluation. A *p*-value below 0.05 was interpreted as significant.

## 3. Results

### 3.1. Clinical Characteristics of the Sample

In total, 38 children and adolescents from 2 to 16 years of age (mean age at first evaluation 8 years and 3 months, 16 females) who underwent presurgical cognitive–behavioral evaluation and/or at least 6 months of follow-up. Two children had specific language and visuo-spatial developmental disorders (see below), and one child was classified as having attentive and emotional disorder. No other neurodevelopmental disorders were present.

Of the patients, 10 had only presurgical evaluation, 10 had only postsurgical evaluation, and 18 had pre- and postsurgical evaluation (mean follow-up time since the surgical intervention: 3 years and 1 month, range 10 months to 11 years). All the subjects were treated with the same protocol of bispectral index-guided anesthesia with hypnotic drugs for the induction and maintenance of sedation (Propofol and Remiphentanyl).

According to the WHO classification system (CNS5), histological diagnoses of the tumors included 29 circumscribed astrocytic gliomas (pilocytic astrocytoma n = 27, pleomorphic xantoastrocytoma n = 2), 8 glioneuronal and neuronal tumors (ganglioglioma n = 4, cerebral neurocytoma n = 1, desmoplastic infantile ganglioglioma n = 1, dysembryoplastic neuroepithelial tumor n = 2) and 1 pediatric-type diffuse low-grade glioma (angiocentric glioma n = 1).

In only eight cases, molecular analyses (BRAF mutation) were performed, and BRAF p.V600 mutation was found in three patients (two with pleomorphic xantoastrocytoma and one with ganglioglioma). These subjects presented a higher age at symptom onset than the whole sample (from 87 to 133 months, mean age 105, SD = 20), also presenting temporal tumors and persistent seizures after surgery. 

Concerning tumor localization, 15 patients presented a cortical hemispheric lesion (temporal n = 11; parietal n = 2; frontal n = 2; circumscribed astrocytic gliomas n = 7; glioneuronal tumors n = 7; pediatric-type diffuse low-grade gliomas n = 1), 7 subjects had tumors located in subcortical structures (thalamic zone n = 6; hypothalamus/pituitary zone n = 1; circumscribed astrocytic gliomas n = 7), and 16 subjects had tumors located in the posterior fossa (cerebellum n = 13; brainstem n = 3; circumscribed astrocytic gliomas n = 15; glioneuronal tumors n = 1). As expected, histological type was related to tumor localization: all the glioneuronal tumors except one were located in cortical structures, while tumors located in the posterior fossa were more often circumscribed astrocytic gliomas. 

Fourteen patients with tumors localized in cortical hemispheres suffered from presurgical seizures (circumscribed astrocytic gliomas n = 7; glioneuronal tumors n = 7). All presented focal seizures, five of them (all glioneuronal tumors) presented an electro-clinical status requiring more than one anti-seizure medication, and seven of them (glioneuronal tumors n = 4 and circumscribed astrocytic gliomas n = 3) were free of disabling seizures after surgery, according to Engel classification of postoperative outcomes [35], although they continued to take anti-epileptic drugs. 

Clinical and demographic variables of the whole sample and of each subsample are shown in Table 1. 

### 3.2. Cognitive Outcome

The pre- and postsurgical cognitive profiles of the three samples are shown in Table 2. All groups presented cognitive abilities within the normal range. One girl presented a specific visuo-perceptive deficit but had verbal abilities in the average range, while another girl presented a language delay but had nonverbal abilities within normal limits. The cognitive profile of the whole sample was characterized by less efficient WM/FD and PS than VIQ and PIQ, with significant differences both in the pre- and postsurgical period (repeated measures ANOVA presurgery F = 4.043, *p* = 0.004, observed power = 0.672; postsurgery F = 4.973, *p* = 0.007, observed power = 0.897). Although the group receiving only postsurgical evaluation presented lower cognitive levels than the other two groups, independent sample *t*-tests did not reach the significance level when separately comparing pre- and postsurgical evaluations (presurgery N = 28, FIQ: test t = −1.593, *p* = 0.062; postsurgery N = 28, FIQ: t = 0.334, *p* = 0.370). In the whole sample, no sex/gender differences were present in cognitive measures (test t from −0.585 to 1.346, *p* from 0.095 to 0.437). Higher age at evaluation was correlated with lower presurgical PS, in particular Symbol search score (rho = −0.669, *p* = 0.006).

#### 3.2.1. Timing and Outcome of Surgery

A longer time between symptoms’ onset and intervention was associated with lower cognitive abilities in FIQ, VIQ, PIQ and WM in the presurgical period (time between symptoms’ onset and intervention and FIQ: rho = −0.636, *p* = 0.001; VIQ: rho = −0.528, *p* = 0.012; PIQ: rho = −0.564, *p* = 0.006; WM: rho = −0–726, *p* = 0.008). A longer period between intervention and postsurgical evaluation was related to lower postsurgical performances in the Similarities and Letter-number sequencing subtests (Similarities: rho = −0.499, *p* = 0.008; Letter-number: rho = −0.864, *p* = 0.001). 

Patients who underwent partial removal of a tumor presented better postsurgical performance in Letter-number sequencing and Abstract reasoning than patients who underwent a complete and more invasive removal of the tumor (Letter-number sequencing: t = −2.093, *p* = 0.033; Abstract reasoning: t = −2.065, *p* = 0.031). No effects of seizure outcome after surgery was present. Figure 1 describes the correlation patterns between cognitive scores and surgical treatment timing.

#### 3.2.2. Clinical Variables

First of all, higher age at symptoms’ onset was related to better FIQ and VIQ in the postsurgical period (FIQ: rho = 0.426, *p* = 0.034; VIQ: rho = 0.456, *p* = 0.022), although the significant correlations did not survive the Bonferroni correction for multiple comparison. 

Lower presurgical cognitive abilities were present in patients with cortical hemispheric tumors and symptomatic seizures when compared to subjects with tumors in other structures (subcortical and posterior fossa); in particular, cortical tumors and preoperative seizures were associated with lower presurgical scores on the FIQ and WM indices and on the Similarities, Information and Digit-span subtests (independent sample *t*-test FIQ: t = 1.926, *p* = 0.033; WM t = 1.898, *p* = 0.043; Similarities t = 1.848, *p* = 0.043; Information t = 1.798, *p* = 0.048; Digit-span t = 2.121, *p* = 0.030). 

Tumor histological type (astrocytic vs. glioneuronal) did not influence pre- or postsurgical cognitive functioning. 

### 3.3. Behavioral Outcome

The pre- and postsurgical behavioral profiles according to the parent-completed CBCL surveys of the three samples are showed in Table 3. All groups presented behavioral and emotional regulation within the normal range without significant differences between internalizing and externalizing problems. 

In the whole sample, no differences in CBCL scores were present according to sex/gender (Test t from −1.094 to 1.020, *p* from 0.285 to 0.457). Higher age at first evaluation was related to higher CBCL Internalizing problems in the postsurgical period (rho = 0.406, *p* = 0.040). 

#### 3.3.1. Timing and Outcome of Surgery

Higher age at intervention was related to greater CBCL Internalizing problems in the postsurgical period (rho = 0.552, *p* = 0.003). On the other hand, a longer time between diagnosis and intervention was related to greater presurgical Externalizing problems (rho = 0.509, *p* = 0.013), and a longer time between symptoms’ onset and intervention was related to greater Externalizing problems in the postsurgical period (rho = 0.046, *p* = 0.019).

Looking at seizure outcomes in the postsurgical period, patients with persistent seizures after surgery presented higher postsurgical Affective and Thought problems when compared to patients free of seizures (Affective: t = −1.920, *p* = 0.039; Thought t = −2.332, *p* = 0.015). Interestingly, patients with persistent postsurgical seizures had higher scores for CBCL Internalizing problems, Withdrawn problems and Thought problems also in the preoperative period (CBCL/Int: t = −1.755, *p* = 0.047; Withdrawn problems: t = −1.910, *p* = 0.034; Thought problems: t = −2.092, *p* = 0.029). 

No effects of partial vs. total surgical removal on pre- or postsurgical behavioral functioning were present. Significant correlations regarding behavioral–emotional functioning are shown in Figure 2.

#### 3.3.2. Clinical Variables

Higher age at symptoms’ onset was related to greater CBCL Internalizing problems in the postsurgical period (rho = 0.608, *p* = 0.002). 

Glioneuronal tumor type was associated with greater presurgical difficulties in CBCL Externalizing, Total problems, Thought problems, Attention Problems and Aggressive behaviors (CBCL/Ext: t = −2.306, *p* = 0.015; CBCL/Tot: t = −1.804, *p* = 0.042; Thought problems: t = −2.326, *p* = 0.019; Attention Problems: t = −2.118, *p* = 0.023; Aggressive Behaviors: t = −2.665, *p* = 0.007). Furthermore, the glioneuronal tumor type was associated with greater postsurgical difficulties in CBCL Internalizing, Externalizing and Total problems (CBCL/Int: t = −1.916, *p* = 0.034; CBCL/Ext: t = −3.529, *p* = 0.001; CBCL/Tot: t = −3.605, *p* = 0.001). Similar results regarding behavioral functioning were obtained when comparing patients with cortical lesions and preoperative seizures (vs. subcortical and posterior fossa localizations). These results were expected because the glioneuronal type was related to cortical localization of the tumor and preoperative seizures, coherently with the histological type. In particular, patients with cortical hemispheric tumors and preoperative seizures showed greater presurgical Sleep problems, higher postsurgical Aggressive, Affective and Withdrawn problems, as well as greater pre- and postsurgical Total problems and Externalizing problems when compared to patients with tumors in other structures (subcortical and posterior fossa) (Sleep problems presurgery: t = −2.234, *p* = 0.026; Aggressive problems postsurgery: t = −2.355, *p* = 0.014; Affective problems postsurgery: t = −3.815, *p* = 0.002; Withdrawn problems postsurgery: t = −1.906, *p* = 0.034; CBCL/Tot presurgery: t = −2.2396, *p* = 0.013; postsurgery: t = −1.909, *p* = 0.034; CBCL/Ext presurgery: t = −1.939, *p* = 0.033; postsurgery: t = −2.796, *p* = 0.005). Notably, when considering the sub-group of patients with cortical tumors and preoperative seizures, the glioneuronal type was still associated with worse pre- and postsurgical behavioral functioning than the astrocytic type, indicating a possible augmented risk for behavioral problems associated with the glioneuronal type regardless of the effect of localization and preoperative seizures (presurgery: CBCL/Ext: t = −3.087, *p* = 0.012; CBCL/Tot: t= −2.528, *p* = 0.030; postsurgery: CBCL/Ext: t = −2.859, *p* = 0.019; CBCL/Tot: t = −3.371, *p* = 0.008).

### 3.4. Effect of Surgery

We analyzed separately the subsample of patients who underwent both pre- and postsurgical assessment, with the aim of evaluating the effect of surgery in predicting cognitive and behavioral outcomes while controlling for the baseline preoperative functioning of the subjects (Figure 3).

A paired samples *t*-test revealed a substantial stability among cognitive abilities after surgery (paired samples *t*-test FIQ T = 0.589, *p* = 0.282; VIQ T = −0.689, *p* = 0.251; PIQ T = −0.281, *p* = 0.391; ML/FD T = 0.552, *p* = 0.30; PS T = −0.283, *p* = 0.393). Moreover, comparisons between pre- and postsurgical CBCL scores did not show significant changes after surgery regarding behavioral and emotional functioning (paired samples *t*-test CBCL/Int T = 0.991, *p* = 0.169; CBCL/Ext T = 0.798, *p* = 0.219; CBCL/Tot T = 1.238, *p* = 0.117). 

A repeated measures ANOVA revealed only univariate effects of glioneuronal tumor type and cortical localization, but no effect of surgery, on general cognitive and behavioral externalizing functioning in this subsample (cortical localization: FIQ F = 6.086, *p* = 0.025, observed power = 0.640; Similarities F = 5.895, *p* = 0.041, observed power = 0.569; glioneuronal type: CBCL/Ext F = 13.007, *p* = 0.003, observed power = 0.918; cortical localization: CBCL/Ext F = 7.829, *p* = 0.015 observed power = 0.735). 

The univariate effects of both variables remained significant when age at symptoms’ onset and age at intervention were added as covariates to the model (glioneuronal type x age at symptoms’ onset: CBCL/Ext F = 14.028, *p* = 0.003, observed power = 0.930; cortical localization x age at symptoms’ onset: FIQ F = 4.446, *p* = 0.050, observed power = 0.497; CBCL/Ext F= 9.995, *p* = 0.009, observed power = 0.820; glioneuronal type x age at intervention: CBCL/Ext F = 12.245, *p* = 0.004, observed power = 0.898; cortical localization x age at intervention: FIQ F = 8.839, *p* = 0.009, observed power = 0.793; CBCL/Ext F = 7.465, *p* = 0.018, observed power = 0.709). 

## 4. Discussion

The present study aimed to describe the cognitive and behavioral outcomes of children and adolescents with isolated low-grade CNS tumors, without other neurological conditions and treated only with surgery, and to identify clinical features associated with intellective and behavioral functioning. In particular, we aimed to investigate the effect of variables still discussed in the literature such as tumor localization, surgical timing and surgical outcome; we also wanted to provide new insight about the effect of low-grade histological subtypes and investigate how clinical and treatment factors may influence behavioral and emotional regulation, which are often neglected in the literature in favor of cognition. 

Our sample of 38 children and adolescents presented cognitive abilities and behavioral functioning in the normal range, in line with the literature [4,13,21]. The cognitive profile is characterized by weaknesses in verbal working memory and processing speed compared to general verbal and nonverbal reasoning, in line with most commonly reported findings in studies about neurocognitive late effects of CNS tumors in childhood [1,2]. Although the patients’ cognitive abilities seem to be quite stable after surgery when considering the pretreatment functioning as a baseline, cognitive efficiency is inversely related to age at evaluation, indicating that patients tested at older ages present worse abilities, as already observed by other research groups [9,12]. Moreover, a longer time between intervention and postsurgical evaluation is related to lower abilities in verbal reasoning and working memory, confirming that long-term follow-up assessments are necessary to monitor the course of neurocognitive problems related to CNS tumors. The present follow-up did not describe worsening of cognitive–behavioral abilities in a relatively small retest time, but the presence of worse functioning in patients tested at older ages and at longer times after surgery supports the idea that neurocognitive sequelae should be monitored in time. Younger age at symptoms’ onset and longer time between symptoms’ onset and intervention are related to worse general cognitive abilities, confirming that low-grade tumors can be still disabling for cognitive development, despite their slow-growing nature; therefore, timely treatment is an important factor influencing the global outcome and daily functioning of patients [15]. Moreover, an earlier onset of symptoms influences a young brain that is in the middle of functional specialization and development, and thus is more susceptible to brain damage and ensuing cognitive impairment [36,37]. On the other hand, when the diagnostic–therapeutic process begins at older ages, this may determine higher levels of emotional and internalizing problems for the child–adolescent, at least in part correlated with greater consciousness and worries in facing the disease and the therapeutic course [2]. 

Concerning the relation between cognitive and behavioral functioning and other clinical variables, we have found that cortical tumors with preoperative seizures and total surgical resection were related to worse cognitive outcomes, and that histological type, cortical tumor and persistent postsurgical seizures were related to higher levels of internalizing and externalizing behavioral problems. The complete and more invasive resection of the tumor was associated with lower working memory and abstract reasoning abilities only in the postsurgical period, while tumor localization in cortical structures influenced cognitive and behavioral regulation both before and after surgical treatment. Interestingly, the presence of persistent postoperative seizures was related to greater postoperative externalizing disorders, but also to greater preoperative affective, thought and sleep problems, suggesting that preoperative behavioral functioning could also be related to postsurgical seizure outcomes, as already described for the cognitive level [38]. 

When focusing on a subsample of children and adolescents who underwent both presurgical evaluation and postsurgical follow-up, thus controlling for the presurgical baseline functioning of the patients, both cognitive abilities and behavioral regulation are found to be substantially stable after surgery and seem to be influenced by clinical variables such as histological type and tumor localization in cortical sites, rather than by surgical treatment per-se.

The present results confirm that, in presence of low-grade CNS tumors occurring during childhood, surgical management is feasible and has favorable outcomes [6,11,13,14], despite challenging and not completely protecting neurocognitive functions and behavioral regulation [7]. Moreover, pretreatment factors are relevant in determining intellective and behavioral sequelae, as already evidenced by some reports [10,13]: in particular, we have confirmed the roles of tumor cortical location and associated preoperative seizures, and we have provided new data about the influence of age at symptoms’ onset and histological low-grade subtype. 

The age at symptoms’ onset is less investigated than other developmental variables such as age at diagnosis and age at intervention, probably because it is more difficult to obtain and to evaluate retrospectively. However, its role is very important, particularly in low-grade tumors that are slow-growing in nature and often present subtle onsets of symptoms, but can still have an effect on brain functioning from the very beginning of their course. As for our results about histological types, cognitive sequelae of glioneuronal tumors have been extensively described, with better courses if freedom from seizures is acquired [39,40]; however, to the best of our knowledge, the present study is the first comparing the behavioral outcomes in different low-grade histological types. Lower IQ and higher occurrence of behavioral problems in children and adolescents with cortical hemispheric tumors have already been described in other samples of children and adolescents with low-grade CNS tumors who have undergone different treatments [4,11,18]. Notably, other reports described weaker functioning in infratentorial than supratentorial tumors [10,20] or failed to find differences in cognitive and neuropsychological outcomes according to tumor localizations [41,42]. In our sample, most cortical tumors were localized in the temporal and frontal lobes, validating the additional risk of working memory difficulties and behavioral dysregulation/externalizing disorders in childhood patients with lesions localized in these brain regions [38,43]. As for the negative impact of complete resection present in our sample, previous literature described larger residual tumor size as predictor of better cognitive functioning in children and adolescents with supratentorial hemispheric tumors [12], as well as lower cognitive abilities in visual processing and processing speed in children and adolescents with complete/subtotal (compared to partial) resection of posterior fossa low-grade tumors [10], but also better verbal working memory in children and adolescents with gross total (compared to subtotal) resection of low-grade tumors in different sites [11]. Further studies are necessary to better analyze the influence of tumor localization and residual tumors; in this framework, collecting baseline presurgical functioning and multiple postsurgical follow-ups is more appropriate to assess the “growth of deficit” phenomenon, i.e., children without deficits in the short-term period after surgery may develop cognitive, social and behavioral deficits years after the acquired injury [12,44]. In fact, also in children treated only with surgery, the effects of brain damage on neurobehavioral functioning during development may be cumulative and more evident in time as such cognitive, social and behavioral functions are expected to mature and be subserved by intact brain networks [12]. 

The present study has some limitations. First of all, the sample size is quite small, and the data about presurgical evaluations are incomplete, leading to low statistical power in the subgroups’ definitions. Larger samples, collected by prospective multicenter studies, are necessary to conduct more complex multivariate analyses to compare different weights of the considered variables in determining cognitive and behavioral outcomes (e.g., disentangling the weight of histological type, cortical localization and preoperative seizures). Notably, when analyzing only the subgroup of patients with cortical tumors and preoperative seizures, the glioneuronal type (vs. the astrocytic type) was still associated with worse pre- and postsurgical behavioral functioning, indicating a possible augmented risk for behavioral problems associated with the glioneuronal type regardless of the effect of localization and seizures. Retrospective collection of cases did not allow the a priori selection of the sample and limited the investigation of some predictors, such as presurgical hydrocephalus, the specific brain area affected by the tumor and the use of antiepileptic drugs, given the lack of available data. Moreover, we cannot exclude possible postoperative cognitive dysfunctions caused by anesthetics, although we can estimate a low risk in our cohort of patients with at least 6 months’ follow-up; in fact, the anesthetics’ effects described in the literature are transitory, observed within 3 to 6 months after surgery [45,46,47]) and negligible in children older than 4 years [46].

Another limitation concerns the sole use of intellective scales as cognitive outcome measure: although less comprehensive than the specific neuropsychological functioning [10,13,14], intellectual scales are the most widely used, being validated in several languages and in several clinical and research contexts, and their use should be mandatory in observational research to guide the definition of clinical trial protocols. Moreover, we did not collect data about environmental variables from a bio-psychosocial perspective [2,48], such as familial and socioeconomic status, school support, rehabilitation treatment and everyday activity. On the other hand, we have provided results about emotional and behavioral sequelae that are often neglected in studies about psychological functioning with CNS tumors in childhood [13,20,38,49].

Furthermore, molecular analyses were performed to date in few cases, and thus it was not possible to investigate more deeply the influence of mutational status in predicting cognitive outcomes. Interestingly, the three BRAF *p*.V600E-mutated patients presented a higher age at symptom onset than the whole sample, also presenting temporal tumor location and persistent seizures after surgery. In particular, in the light of the present results of the glioneuronal subtype, it will be interesting to figure out the prognostic significance of a specific molecular status on the cognitive, neuropsychological and behavioral features [7] that could be analyzed in further studies.

In conclusion, the present study described cognitive and behavioral sequelae of pediatric low-grade CNS tumors in children and adolescents treated only with surgery and without premorbid risk factors, also considering the preoperative cognitive and behavioral functioning as a baseline. We have provided the first comparisons between different histological subtypes within low-grade tumors and evidence about the influence of the glioneuronal subtype (and clinical associated variables) on patients’ behavioral-emotional regulation, which is still very poorly investigated. The obtained results suggest that cognitive and behavioral functioning is related to pretreatment variables (age at symptoms’ onset, histological type, cortical tumor location), timing of surgery and seizure control after surgery, and is stable when controlling for a preoperative cognitive and behavioral baseline. Younger age at onset is confirmed as a particularly vulnerable period in determining cognitive sequelae, while children and adolescents at older ages or at longer postsurgical follow-ups are at higher risk for developing behavioral disturbances. Timely treatment is an important factor influencing the global outcome and quality of life of the patients. Preoperative and regular postsurgical cognitive and behavioral assessments, also several years after surgery, should be included in standard clinical practices.

## Figures and Tables

**Figure 1 diagnostics-13-01568-f001:**
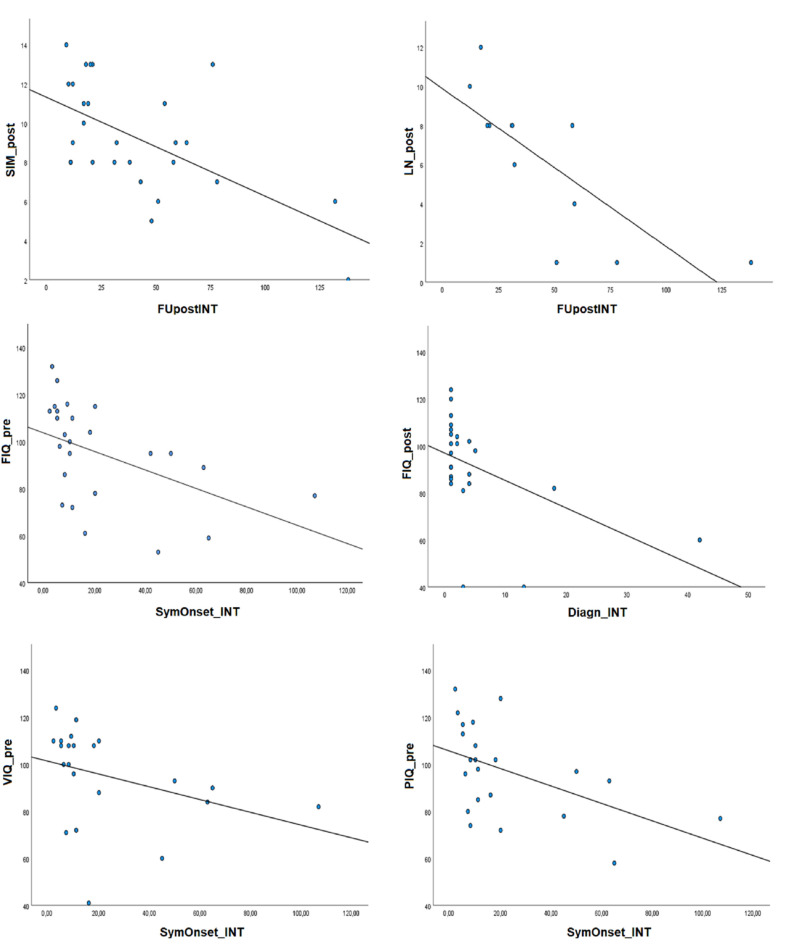
Correlations between cognitive functioning and surgical treatment timing. Variables about treatment timing are expressed in months; cognitive variables are expressed in standardized weighted scores and quotients. FUpostINT: tie between intervention and postsurgical evaluation, Diagn_INT: time between diagnosis and intervention, SympOnset_INT: time between symptom onset and intervention, SIM_post: Similarities postsurgery, LN_post: Letter-number sequencing postsurgery, FIQ: Full intelligence quotient; VIQ: verbal intelligence quotient, PIQ: performance intelligence quotient.

**Figure 2 diagnostics-13-01568-f002:**
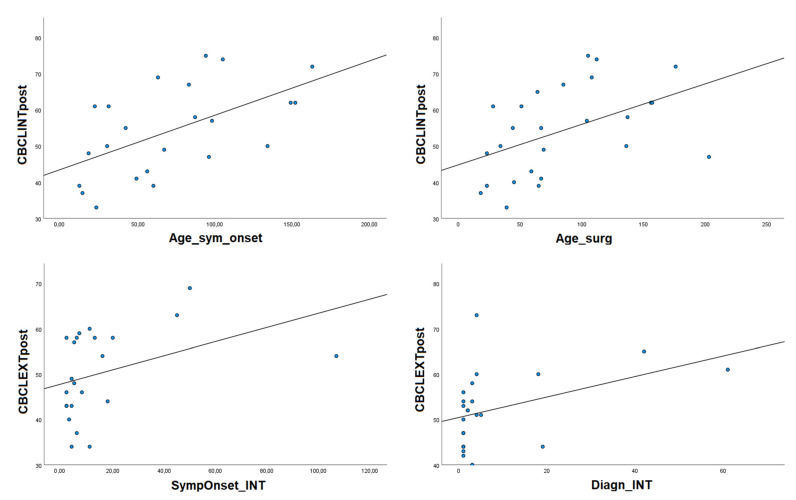
Correlations between behavioral functioning, symptoms’ onset and surgical treatment timing. Variables about diagnostic and treatment timing are expressed in months; behavioral variables are expressed in standardized T-scores. Age_sym_onset: age at symptoms’ onset, Diagn_INT: time between diagnosis and intervention; SympOnset_INT: time between symptom onset and intervention, CBCLInt: Child Behavior Checklist Internalizing; CBCLExt: Child Behavior Checklist Externalizing.

**Figure 3 diagnostics-13-01568-f003:**
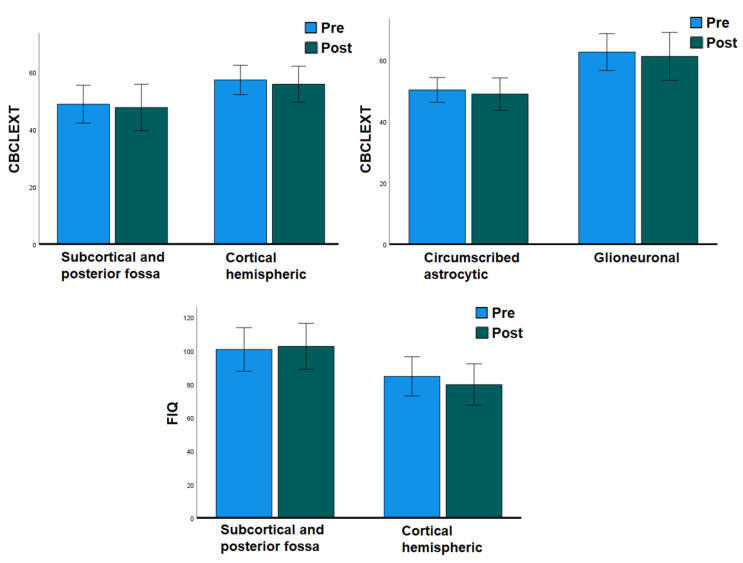
Pre- and postsurgical cognitive and behavioral functioning according to tumor type and localization. Cognitive variables are expressed in standardized quotients; behavioral variables are expressed in standardized T-scores.

**Table 1 diagnostics-13-01568-t001:** Clinical and demographic description of the sample.

Clinical and Demographic Variables	Total Sample (n = 38)	Pre-post Follow-Up (*n* = 18)	Only Presurgical Evaluation (*n* = 10)	Only Postsurgical Evaluation (*n* = 10)
Age at first evaluation (mean, SD, range) (months)	99 ± 46 (23–199)	90 ± 50 (23–199)	95 ± 34 (59–159)	119 ± 50 (64–198)
Age at symptoms’ onset (months)	71 ± 43 (12–163)	72 ± 46 (12–152)	79 ± 36 (12–133)	62 ± 48 (14–163)
Age at surgery (months)	85 ± 47 (18–203)	91 ± 51 (23–203)	98 ± 37 (59–161)	66 ± 47 (18–176)
Sex (female, %)	15 (39.5)	9 (50)	5 (50)	1 (10)
Histological type (*n*, %)				
- Circumscribed astrocytic gliomas	29 (76.3)	12 (66.7)	8 (90)	9 (90)
- Glioneuronal and neuronal tumors	8 (15.8)	6 (27.8)	1 (10)	1 (10)
- Pediatric-type diffuse low-grade gliomas	1 (3.8)	0	1 (1)	0
Localization of tumor (*n*, %)				
Cortical hemispheric	15 (39.5)	10 (55.6)	4 (40)	1 (10)
- Temporal	11 (28.9)	7 (38.9)	3 (30)	1 (10)
- Parietal	2 (5.3)	1 (5.6)	1 (10)	0
- Frontal	2 (5.3)	2 (11.1)	0	0
Sub-cortical	7 (18.4)	4 (22.3)	3 (30)	0
- Thalamic zone	6 (15.8)	3 (16.7)	3 (30)	0
- Hypothalamus/Pituitary zone	1 (2.6)	1 (5.6)	0	0
Posterior Fossa	16 (42.2)	3 (16.7)	4 (40)	9 (90)
- Cerebellum, hemisphere	6 (15.8)	2 (11.1)	2 (20)	2 (20)
- Cerebellum, vermis	5 (13.2)	1 (5.6)	0	4 (40)
- Cerebellum, hemisphere and vermis	2 (5.3)	0	2 (20)	0
- Brainstem	3 (7.9)	0	0	3 (30)
Lateralization of tumor				
- Left	15 (39.5)	6 (33.3)	7 (70)	2 (20)
- Right	17 (44.7)	12 (66.7)	3 (30)	2 (20)
- Median/bilateral	6 (15.8)	0	0	6 (60)
Surgical removal				
- Complete	22 (57.9)	10 (55.6)	5 (50)	7 (70)
- Partial	16 (42.1)	8 (44.4)	5 (50)	3 (30)
Seizures outcome after surgery *				
- Seizures	7 (18.4)	3 (22.2)	3 (30)	1 (10)
- Free of seizures	31 (81.6)	15 (77.8)	7 (70)	9 (90)
Hydrocephalus				
- Yes	6 (15.8)	2 (11.1)	2 (20)	2 (20)
- No	32 (84.2)	16 (88.9)	8 (80)	8 (80)

Legend: SD: standard deviation; * Seizures outcome after surgery was recorded using the Engel classification system [33].

**Table 2 diagnostics-13-01568-t002:** Pre- and post-treatment cognitive profile of patients treated only with surgery for low-grade brain tumors.

Cognitive Functioning	Pre-Post Group	Only Presurgical Group	Only Postsurgical Group
Presurgery			
- Full IQ	91.7 ± 18.7 (53–115)	104.7 ± 22.1 (59–132)	n.a.
- Verbal IQ	91.2 ± 21.1 (41–119)	104.3 ± 13.2 (84–124)	n.a.
- Performance IQ	94.2 ± 16.6 (72–132)	105.3 ± 22.1 (58–128)	n.a.
- WM/FD	87.4 ± 19.9 (55–110)	94.0 ± 26.7 (58–133)	n.a
- PS	82.6 ± 25.5 (50–123)	96.1 ± 20.9 (65–129)	n.a.
Postsurgery			
- Full IQ	89.8 ± 21.3 (40–124)	n.a.	87.0 ± 21.8 (40–109)
- Verbal IQ	93.7 ± 16.7 (64–120)	n.a.	89.6 ± 17.3 (52–113)
- Nonverbal IQ	96.6 ± 23.5 (45–132)	n.a.	88.3 ± 24.3 (41–112)
- WM/FD	86.5 ± 19.8 (46–118)	n.a.	83.8 ± 22.7 (46–106)
- PS	82.4 ± 16 (47–106)	n.a.	85.7 ± 31.3 (47–123)

Legend: IQ: intelligence quotient; WM: working memory; FD: freedom from distractibility; PS: processing speed; n.a.: not available.

**Table 3 diagnostics-13-01568-t003:** Pre- and post-treatment behavioral profiles of patients treated only with surgery for low-grade brain tumors.

Behavioral Functioning	Pre-Post Group	Only Presurgical Group	Only Postsurgical Group
Presurgery			
- CBCL/Int	56.1 ± 15.3 (33–94)	56.5 ± 5.8 (51–65)	n.a.
- CBCL/Ext	53.9 ± 8.8 (40–73)	50.3 ± 6.8 (43–61)	n.a.
- CBCL/Total	56.4 ± 12.1 (36–83)	52.6 ± 4.1 (48–69)	n.a.
Postsurgery			
- CBCL/Int	55.7 ± 12.9 (33–75)	n.a.	52.8 ± 11.5 (37–72)
- CBCL/Ext	53.8 ± 9.3 (34–69)	n.a.	45.6 ± 9.3 (34–58)
- CBCL/Total	56.3 ± 11.7 (34–72)	n.a.	49.3 ± 11.0 (33–70)

Legend: CBCL: Child Behavior Checklist; Int: Internalizing disorders; Ext: Externalizing disorders; n.a.: not available.

## Data Availability

The data presented in this study are available on request from the corresponding author.

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
