# Peer review of "Cognitive and Behavioral Outcome of Pediatric Low-Grade Central Nervous System Tumors Treated Only with Surgery: A Single Center Experience"

_diagnostics, 2023, doi:10.3390/diagnostics13091568_

Round 1
Reviewer 1 Report
The paper describes the cognitive and behavioral outcome of children and adolescents with low grade CNS tumors, a single center´s experience. The topic is interesting and deserves attention as long-time follow-up of low grade tumors is important. However, there are some concerns that should be addressed.
1. Abstract, line 24. “We selected” – were some patients left out? the word select seems inappropriate, please rephrase, “included” is an alternative.
2. Abstract, line 26-27. “…. and being treated with adjuvant therapies”. Can be deleted, it is already stated above that the cohort only involves patients treated exclusively with surgical intervention.
3. The introduction is somewhat lengthy, it would be helpful if the introduction could be shortened and a little more concise.
4. End of introduction, line 105-109. The aim differs somewhat from the aim that is stated in the abstract and beginning of the discussion section. Given the presented data in the paper the stated aim in the beginning of the discussions section seems more appropriate: “The present study aimed to describe the cognitive and behavioral outcome of children with low-grade CNS tumors treated only with surgery and to identify clinical features associated with intellective and behavioral functioning”.
Aiming to better clarify the effect of surgical treatment can be a second aim, suggestion: “We also wanted to clarify the effect of surgical treatment….”.
5. Introduction, line 41 – CNS abbreviation should be defined.
6. Introduction, line 48 – consider using “chemotherapy” instead of “pharmacological”. Change references from (8,5) to (5,8).
7. Introduction, line 48-52. Consider rephrasing – signs and other specific symptoms are also common in children with low grade tumors, the difference being they occur at a later stage compared to patients with high grade tumors. For instance, children with infratentorial pilocytic astrocytomas commonly present with symptoms related to high intracranial pressure.
Also, increased intracranial pressure cause symptoms, but is not a symptom in itself.
8. Introduction, line 57-60. This meaning is unclear, please clarify. Also, it is stated that most patients are only tested at one time-point, is this a reflection of local traditions? References supporting this statement? In many centers repeated testings post treatment has been established as a routine.
9. Introduction, line 68-69. Suggestion, rephrase to “Children receiving adjuvant chemo- and/or radiotherapy….”
10. Material and Methods, line 116. Include information of patient ages already here.
11. Material and Methods, line 118. Selected, please rephrase.
12. Material and Methods, line 121. NF1, abbreviation should be defined.
13. Material and Methods, regarding the exclusion criteria. Was neurodevelopmental disorders an exclusion criteria as well? ADHD, ADD, atypical autism etc.
14. Material and Methods, line 124-130. Is this information needed here or could it be integrated in section 2.2 and 2.3?
15. Results, 3.1 line 242-243. “who underwent pre-surgical cognitive and/or behavioral evaluation and at least six months’ post-surgical follow-up” is this meaning correct? Shouldn´t it state “pre-surgical cognitive-behavioral evaluation and/or at least six months follow-up” instead?
16. Result, line 244-246. Use “patients” instead of “children” as it involves both children and adolescents.
17. Results, line 288-289. No sex/gender differences in behavioral measures can be put in section 3.3 were these results are presented.
18. Results, Figure 1. Text on x- and y-axis should be enlarged. It is difficult to read. Moreover, x- and y-axis are missing labels, should be written out although it says in the figure caption. It would also be easier to read the results if the each abbreviation would have its own row (instead of being written after each other).
· Sim_post = Similarities post-surgery
· LN_post = Letter-number sequencing post-surgery
· FIQ = Full intelligence quotient
· VIQ = verbal intelligence quotient
· Etc.
19. Results, Figure 2. Same as for Figure 2, see above.
20. Results, Figure 3. Small text, please enlarge.
21. Results, Table 1. Sub-cortical, N should be 7 (not 23). Percentage 18 (not 60.5). Moreover, posterior fossa should read N=16 (corresponding to 42%).
22. Discussion, line 455. “…. older patients are more likely to present worse abilities, as already observed by other research groups” – can the authors elaborate on this? Is it because of a higher frequency of cortical tumors causing seizures/epilepsy?
23. Discussion, line 456-459. “Moreover, longer time between intervention and post-surgical evaluation is related to lower abilities in verbal reasoning and working memory, confirming that long term follow-up assessments are necessary to monitor the course of neurocognitive problems related to CNS tumors”. Is this statement supported by the presented data/results? Also, do the authors have any thoughts on why this is the case? What are the possible explanations behind “growth of deficit” phenomenon regarding cognitive problems in patients treated only with surgery?
24. The authors state both in the abstract, results, discussion section as well as concluding remarks that a cortical tumor location affects both cognitive and behavioral functioning. 14 out 15 patients with a cortical tumor had an epilepsy (presurgical seizures), and were most likely treated with antiepileptic medication. Moreover, 8 patients had a glioneuronal tumor, which in many cases is known to be associated with a severe or even medical refractory epilepsy. What was the preoperative seizure frequency and seizure severity in this cohort? Is it these circumstances, rather than the tumor location itself, that affect the findings? Please elaborate on this.
Author Response
Response to Reviewer 1 Comments
General comment: The paper describes the cognitive and behavioral outcome of children and adolescents with low grade CNS tumors, a single center´s experience. The topic is interesting and deserves attention as long-time follow-up of low grade tumors is important. However, there are some concerns that should be addressed.
Response: Thank you very much for acknowledging the strength of the paper and the importance of the topic, please find below our point by point answers to your concerns and the corresponding lines in the current version of the manuscript.
Point 1: Abstract, line 24. “We selected” – were some patients left out? the word select seems inappropriate, please rephrase, “included” is an alternative.
Response 1: we agree with the observation of the reviewer, the patients were retrospectively selected, thus we changed accordingly in “included” (line 27).
Point 2: Abstract, line 26-27. “…. and being treated with adjuvant therapies”. Can be deleted, it is already stated above that the cohort only involves patients treated exclusively with surgical intervention.
Response 2: we changed accordingly deleting the phrase (line 30).
Point 3: The introduction is somewhat lengthy, it would be helpful if the introduction could be shortened and a little more concise.
Response 3: we appreciate this suggestion that could ameliorate the quality of the paper. We have shortened the introduction trying to be more concise and going straight to the point of what is new in the present investigation and what are the specific aims of the study, integrating also the observations of another reviewer (Reviewer 3).
Point 4: End of introduction, line 105-109. The aim differs somewhat from the aim that is stated in the abstract and beginning of the discussion section. Given the presented data in the paper the stated aim in the beginning of the discussions section seems more appropriate: “The present study aimed to describe the cognitive and behavioral outcome of children with low-grade CNS tumors treated only with surgery and to identify clinical features associated with intellective and behavioral functioning”. Aiming to better clarify the effect of surgical treatment can be a second aim, suggestion: “We also wanted to clarify the effect of surgical treatment….”.
Response 4: we appreciate this suggestion and we corrected the aim of the study at the end of the Introduction accordingly (lines 102-114), also specifying the investigation of the effect of surgical treatment as a second aim (lines 111-114). Please notice that we have been a little more concise about the investigated variables in order to follow the observation of another reviewer which found the scope of the study too generic (Reviewer 3).
Point 5: Introduction, line 41 – CNS abbreviation should be defined.
Response 5: sorry for missing the definition of this abbreviation, we have modified accordingly (line 45).
Point 6: Introduction, line 48 – consider using “chemotherapy” instead of “pharmacological”. Change references from (8,5) to (5,8).
Response 6: we have modified accordingly (line 53).
Point 7: Introduction, line 48-52. Consider rephrasing – signs and other specific symptoms are also common in children with low grade tumors, the difference being they occur at a later stage compared to patients with high grade tumors. For instance, children with infratentorial pilocytic astrocytomas commonly present with symptoms related to high intracranial pressure. Also, increased intracranial pressure cause symptoms, but is not a symptom in itself.
Response 7: thank you for this very punctual observation. We have rephrased evidencing the different onset timing of low-grade tumors symptoms and referring to symptoms caused by intracranial pressure (not intracranial pressure itself) (lines 53-57).
Point 8: Introduction, line 57-60. This meaning is unclear, please clarify. Also, it is stated that most patients are only tested at one time-point, is this a reflection of local traditions? References supporting this statement? In many centers repeated testings post treatment has been established as a routine
Response 8: we have rephrased to clarify the expressed concepts and providing references (lines 62-68). We agree with the reviewer that longitudinal post-surgical testing is guaranteed as a routine in several centers, but we believe that longitudinal testing still has a drop out risk limiting data collection for research; moreover, only post-surgical evaluation does not account for possible pre-existing conditions.
Point 9: Introduction, line 68-69. Suggestion, rephrase to “Children receiving adjuvant chemo- and/or radiotherapy….”
Response 9: this sentence has been deleted in the new version of the introduction.
Point 10: Material and Methods, line 116. Include information of patient ages already here.
Response 10: we have modified accordingly including information about patients age ranges (line 121).
Point 11: Material and Methods, line 118. Selected, please rephrase.
Response 11: we have rephrased accordingly (line 117).
Point 12: Material and Methods, line 121. NF1, abbreviation should be defined..
Response 12: we have defined NF1 abbreviation (line 124).
Point 13: Material and Methods, regarding the exclusion criteria. Was neurodevelopmental disorders an exclusion criteria as well? ADHD, ADD, atypical autism etc.
Response 13: We decided not to apriori exclude children with neurodevelopmental disorders because this could have determined missing some important behavioral variability in our sample. However, in the final sample, we verified that none had ASD or intellectual disability, two children had pre-surgical specific language and visuo-spatial disorders as investigated by standardized tests and one child was classified with attention and emotional dysregulation. We better specified this information in the results (lines 251-253).
Point 14: Material and Methods, line 124-130. Is this information needed here or could it be integrated in section 2.2 and 2.3?
Response 14: we agree that this information is redundant here, we have integrated it in sections 2.2. and 2.3 (lines 152-153).
Point 15: Results, 3.1 line 242-243. “who underwent pre-surgical cognitive and/or behavioral evaluation and at least six months’ post-surgical follow-up” is this meaning correct? Shouldn´t it state “pre-surgical cognitive-behavioral evaluation and/or at least six months follow-up” instead?
Response 15: thank you for rephrasing, we have modified accordingly (250-251).
Point 16: Result, line 244-246. Use “patients” instead of “children” as it involves both children and adolescents.
Response 16: we think that this comment may be useful for the entire manuscript thus we changed accordingly using “children and adolescents” or “patients” throughout the manuscript.
Point 17: Results, line 288-289. No sex/gender differences in behavioral measures can be put in section 3.3 were these results are presented.
Response 17: thank you for this observation, this was our mistake, in fact sex/gender differences in behavioral measures were already present in section 3.3.
Point 18: Results, Figure 1. Text on x- and y-axis should be enlarged. It is difficult to read. Moreover, x- and y-axis are missing labels, should be written out although it says in the figure caption. It would also be easier to read the results if the each abbreviation would have its own row (instead of being written after each other).
- Sim_post = Similarities post-surgery
- LN_post = Letter-number sequencing post-surgery
- FIQ = Full intelligence quotient
- VIQ = verbal intelligence quotient
- Etc.
Point 19. Results, Figure 2. Same as for Figure 2, see above.
Response 18 and 19: we have enlarged the text in both Figure 1 and 2 and we have written each abbreviation in a row (in captions) (lines 602-611). We are not sure of what the reviewer means with “x- and y-axis are missing labels”.
Point 20. Results, Figure 3. Small text, please enlarge.
Response 20: we modified Figure 3 accordingly.
Point 21. Results, Table 1. Sub-cortical, N should be 7 (not 23). Percentage 18 (not 60.5). Moreover, posterior fossa should read N=16 (corresponding to 42%).
Response 21: thank you for this observation, this is a mistake, we corrected Table 1 accordingly.
Point 22. Discussion, line 455. “…. older patients are more likely to present worse abilities, as already observed by other research groups” – can the authors elaborate on this? Is it because of a higher frequency of cortical tumors causing seizures/epilepsy?
Point 23. Discussion, line 456-459. “Moreover, longer time between intervention and post-surgical evaluation is related to lower abilities in verbal reasoning and working memory, confirming that long term follow-up assessments are necessary to monitor the course of neurocognitive problems related to CNS tumors”. Is this statement supported by the presented data/results? Also, do the authors have any thoughts on why this is the case? What are the possible explanations behind “growth of deficit” phenomenon regarding cognitive problems in patients treated only with surgery?
Response 22 and 23: We answer to comments 22 and 23 together because they both refer to results supporting the growth of deficit phenomenon. We agree with the reviewer about the fact that the present follow-up did not describe worsening of cognitive-behavioral abilities in a relatively small retest time, but the presence of worse functioning in patients tested at older age and at longer time after surgery, support the idea that the neurocognitive sequelae should be monitored in time. We have clarified this idea and we also added possible explanation of the growth of deficit phenomenon in patients treated only with surgery (lines 447-450; 516-520).
Point 24. The authors state both in the abstract, results, discussion section as well as concluding remarks that a cortical tumor location affects both cognitive and behavioral functioning. 14 out 15 patients with a cortical tumor had an epilepsy (presurgical seizures), and were most likely treated with antiepileptic medication. Moreover, 8 patients had a glioneuronal tumor, which in many cases is known to be associated with a severe or even medical refractory epilepsy. What was the preoperative seizure frequency and seizure severity in this cohort? Is it these circumstances, rather than the tumor location itself, that affect the findings? Please elaborate on this.
Response 24: thank you for this observation; this is a very important point. We agree with all the observations of the reviewer: preoperative seizure severity could have influenced the cognitive-behavioral outcome, the presence of preoperative seizure is associated with cortical localization (as we described), and we could not account for the influence of antiepileptic drugs (as we disclosed in the limitation section).
In our sample, all the glioneuronal tumors except one were cortical, but not all the cortical tumors leading to seizures were glioneuronal; moreover, when considering only the sub-group of patients with cortical tumors and preoperative seizures, glioneuronal type was still associated with worse pre- and post-surgical behavioral functioning than astrocytic type, indicating a possible augmented risk for behavioral problems associated to glioneuronal type regardless the effect of localization and preoperative seizures.
We are also aware of the limited power of our sample that limits the possibility to disentangle the role of all the associated variables (in particular, histology, localization, and seizures). The limited power has been emphasized by another reviewer (Reviewer 3), who suggested to “emphasize certain aspects/variables rather than trying to account for everything”. This is the reason why in the present version of the manuscript we elaborated more on specific variables that, to our knowledge, are poorly discussed in the literature (histology, age at symptoms onset) or provided conflicting results (localization, surgical residual and seizures control after intervention).
To follow the suggestion of Reviewer 1, we have done the following changes to the manuscript:
1) we better elaborated this argument by adding pre-operative seizures severity to the sample description (lines 281-283)
2) in the abstract, results, and discussion, we always presented localization and seizures as associated variables
3) we emphasized the possible augmented risk for behavioral problems associated to glioneuronal type regardless the effect of localization and preoperative seizures also in the discussion (lines 526-530).
Please see the revised manuscript.
Reviewer 2 Report
Well written and good study. This manuscripts presents the presurgical cognitive performance of children with low grade CNS tumour. Some of the result of the study should be written in the abstract, such as median age, gender, types of low grade CNS tumours, cognitive outcome, etc. The limitations of the were incomplete data (only 18 out of 38 subjects with complete data, pre and post surgery assessment). The authors may suggest to perform prospective multicentre study with a bigger cohort to verify the findings.
Author Response
Response to Reviewer 2 Comments
Point 1: Well written and good study. This manuscripts presents the presurgical cognitive performance of children with low grade CNS tumour. Some of the result of the study should be written in the abstract, such as median age, gender, types of low grade CNS tumours, cognitive outcome, etc. The limitations of the were incomplete data (only 18 out of 38 subjects with complete data, pre and post surgery assessment). The authors may suggest to perform prospective multicentre study with a bigger cohort to verify the findings.
Response 1: Thank you for acknowledging the strength of the study and for your very positive comments.
In order to follow the suggestion of the reviewer, we have written in the abstract some descriptive data of the sample (lines 27-28, 30-33).
Moreover, we have suggested prospective multicentre study to overcome the limitation of the present single centre report (lines 522-523).
Please see the revised manuscript.
Reviewer 3 Report
This paper provides descriptive data from a single center examining cognitive and behavioral outcomes from pediatric low-grade central nervous system tumors. The authors include in their aims examination of factors including tumor location, tumor histology, age at symptom onset, age at diagnosis and intervention, extent of surgery, and seizures as potential predictors of outcome.
It would be helpful to report the number of children and adolescents who were excluded from this retrospective study. It would be interesting to see how many were excluded simply because they did not have an evaluation versus those who were excluded because of one of the exclusion criteria. There is potential for selection bias in the sample.
Table 1 is referenced on page 6 (line 273) but is not presented until page 10. Similarly, table 2 is not presented until several pages after it is referenced in lines 276-277.
The results overall are very difficult to read. There is a lack of thematic presentation in favor of serial details of specific results. This reads more like a list of results rather than a coherent, thematic investigation.
This approach leads to similar difficulty following the discussion section. The paper overall lacks a coherent aim aside from broadly describing cognitive and behavioral outcomes from pediatric low-grade CNS tumors in a single center.
Because the analyses lack of power due to so many subgroups, the authors may wish to revisit their data with a more well-developed aim emphasizing certain aspects/variables associated with pediatric low-grade glioma rather than trying to account for everything.
Because of the small sample size and the wide diversity of tests used (e.g., so many versions of Wechsler tests which can vary from one to another and are not considered equivalent) it is hard to make sense of what is “learned” from this study. It seems the authors confirm a few findings that have already been demonstrated, but it is not clear what we are learning that is new. Again, I encourage the authors to consider a more targeted analytic plan. This would also help to provide greater focus to the introduction section of the paper.
The paper overall would benefit from English language editing. There are a number of unusual word usages and phrasings that could be cleaned up and would provide greater clarity.
Author Response
Response to Reviewer 3 Comments
Point 1: This paper provides descriptive data from a single center examining cognitive and behavioral outcomes from pediatric low-grade central nervous system tumors. The authors include in their aims examination of factors including tumor location, tumor histology, age at symptom onset, age at diagnosis and intervention, extent of surgery, and seizures as potential predictors of outcome.
It would be helpful to report the number of children and adolescents who were excluded from this retrospective study. It would be interesting to see how many were excluded simply because they did not have an evaluation versus those who were excluded because of one of the exclusion criteria. There is potential for selection bias in the sample.
Response 1: thank you for this suggestion very useful to clarify the retrospective exclusion of the subjects. We have provided this information in the Methods – Participant section (lines 127-134).
Point 2: Table 1 is referenced on page 6 (line 273) but is not presented until page 10. Similarly, table 2 is not presented until several pages after it is referenced in lines 276-277.
Response 2: : to facilitate the reading of results we moved each table after the corresponding paragraph (Table 1 after 3.1 Clinical characteristics of the sample, Table 2 after 3.2 Cognitive outcome and Table 3 after 3.3 Behavioral outcome).
Point 3: The results overall are very difficult to read. There is a lack of thematic presentation in favor of serial details of specific results. This reads more like a list of results rather than a coherent, thematic investigation.
Response 3: in order to follow this useful suggestion of reviewer 3 and better clarify the thematic presentation of results, we divided the general domains “Cognitive” and “Behavioral” outcomes in specific sub-paragraphs, discussing separately a) surgical timing and outcome (both residual and seizures) b) clinical features (age at symptoms onset, histology, localization). Moreover, we did not consider the results about diagnostic timing and evaluation timing that are less original, focusing more on age at symptoms’ onset, histology, localization/seizures and variables related to the timing and outcome of surgery. Finally, we presented and discussed the localization and preoperative seizures as associated variables because they are distributed in the same way (14 out of 15 children with cortical tumors presented symptomatic preoperative seizures).
Point 4: This approach leads to similar difficulty following the discussion section. The paper overall lacks a coherent aim aside from broadly describing cognitive and behavioral outcomes from pediatric low-grade CNS tumors in a single center.
Response 4: thank you for allowing us to clarify the aim of the present study and what is original, we probably did not stress these points enough in the introduction and the discussion. We were interested in describing the functional outcomes of the surgical treatment in low-grade tumors during childhood by a retrospective selection of the children referred to our Institute considering the most important variables described in the literature (localization, seizures, surgical residual), but also variables still not investigated (histological sub-type, age at symptoms onset). The role of histological low-grade sub-type is still not studied (as we discussed, this is the first study, to our knowledge, comparing the behavioral outcomes in different low-grade histological types); moreover, the role of localization, seizures, and tumour residual, is still not clear and may be related to confounding factors such as studying together low and high-grade tumors or including children with other neurological conditions. Finally, our study has the strenght of providing data about preoperative behavioural and emotional functioning and postoperative follow—up of the same subjects (although in a relatively small subsample) that are very scarce in the literature. In this revised version of the manuscript, we have stressed these aims and consequently the strength of the present study in the introduction (line 108-114) and the discussion (line 431-435, 483-494, 560-563).
Point 5: Because the analyses lack of power due to so many subgroups, the authors may wish to revisit their data with a more well-developed aim emphasizing certain aspects/variables associated with pediatric low-grade glioma rather than trying to account for everything.
Response 5: we are aware of the limited power of our analysis, as often revealed in retrospective studies on highly selected samples, and we have acknowledged this issue in the limitation section.
In order to follow the suggestion of the reviewer we decided to emphasize the role of histology, localization/seizures, age of symptoms onset, surgical timing and surgical outcome (seizures and tumor residual). We still think that is valuable to study more than only one or two variables because 1) their role on the cognitive functioning and, above all, on behavioural functioning, is still not clear and 2) because study with highly selected samples are still scarce and the present result may pave the way for future prospective multicentre and more powerful studies (as reported in the limitation section and suggested by another reviewer).
Point 6: Because of the small sample size and the wide diversity of tests used (e.g., so many versions of Wechsler tests which can vary from one to another and are not considered equivalent) it is hard to make sense of what is “learned” from this study. It seems the authors confirm a few findings that have already been demonstrated, but it is not clear what we are learning that is new. Again, I encourage the authors to consider a more targeted analytic plan. This would also help to provide greater focus to the introduction section of the paper.
Response 6: we hope that this comment may be solved by the answer to the previous raised points about results and discussion, and that it could be now clearer what, in our opinion, is original from this paper.
Point 7: The paper overall would benefit from English language editing. There are a number of unusual word usages and phrasings that could be cleaned up and would provide greater clarity.
Response 7: : the manuscript has been checked by a bilingual English-speaking colleague, but we will also ask for the MDPI service for English editing to provide greater clarity to the final version of the paper.
Please see the revised manuscript.
Reviewer 4 Report
Dear authors
I find your article interesting but please include some data about type of anesthesia, drugs used as there are data about cognitive dysfunction caused by anesthetics (primary experimental) in pediatric and advanced age (POCD).
Best regards
Author Response
Response to Reviewer 4 Comments
Point 1: Dear authors, I find your article interesting but please include some data about type of anesthesia, drugs used as there are data about cognitive dysfunction caused by anesthetics (primary experimental) in pediatric and advanced age (POCD).
Response 1: thank you for acknowledging the strengths of the study and for your positive comment. We added data about the type of anesthesia used in the method section (lines 257-259) and commented on it in the discussion (lines 534-538). All the subjects assumed the same protocol of bispectral index-guided anesthesia with hypnotic drugs for induction and maintenance of sedation (Propofol and Remiphentanyl). In general, we can estimate a low risk for postoperative cognitive dysfunction in our cohort because we selected patients with at least 6 months neuropsychological follow-up, while literature reports mainly found negligible (Hansen, 2015 https://doi.org/10.1111/pan.12548) to transitory (Kotekar et al., 2018 https://doi.org/10.2147/CIA.S133896; Aun et al., 2016 https://doi.org/10.1097/MD.0000000000003250) effects (mainly within 3 to 6 months after surgery), and lower effects particularly if the age of children is over 4 years (Aun et al., 2016).
Please see the revised manuscript.
Round 2
Reviewer 3 Report
This version is much improved and much clearer. I appreciate the authors' responses to the points made in the initial review.